# Trends in admission, resource use and outcomes among elderly patients admitted to an intensive care unit in China

Xiaohui Zhu[1], Meiping Wang[2], Yawei Guo[1], Rong Sun[3], Li Jiang [2]*

1 Emergency Department, Xuanwu Hospital of Capital Medical University, Beijing, China, 2 Department of Critical Care Medicine, Xuanwu Hospital of Capital Medical University, Beijing, China, 3 Department of Critical Care Medicine, Shijitan Hospital of Capital Medical University, Beijing, China

* jiangli@ccmu.edu.cn

## Abstract

### Background

Population aging, coupled with improvements in healthcare, may influence ICU admission trends and care practices among critically ill elderly patients (≥80 years), longitudinal data evaluating this remain limited in China. Our aim was to analyze and compare trends in ICU admissions, clinical outcomes, and resource use among critically ill elderly patients (≥80 years), in comparison with older (65–79 years) and younger (16–64 years) cohorts.

### Methods

We retrospectively analyzed ICU patients aged ≥16 years admitted to a tertiary referral hospital in China from January 2014 to December 2021. A total of 31,535 patients were categorized into three age groups: ≥80 years (11.5%), 65–79 years (30.7%), and <65 years (57.8%).

### Results

ICU admission rates for elderly patients declined significantly from 13.2% in 2014 to 9.0% in 2021 (p < 0.001, relative decrease 31.8%), particularly among nonsurgical admissions. Elderly patients had higher comorbidities, greater disease severity scores, but lower daily average TISS-28 scores compared to younger cohorts. They were more likely to receive inotropic/vasopressor support and nutritional interventions, had higher blood loss due to frequent laboratory testing, and required more red blood cell transfusions. However, they were less likely to undergo invasive ventilation. The proportion of elderly patients requiring invasive ventilation decreased significantly over the study period. Despite a higher hospital mortality rate (elderly 14.1%, older 6.1%, younger 3.1%; p < 0.001), elderly patients demonstrated a more

**Data availability statement:** All relevant data are within the paper and its Supporting information files. Chinese data privacy regulations and the ethical approval (Xuanwu Hospital Human Research Ethics Committee, approval number 2024(047)-001) prohibit public sharing of individual-level data. Anonymized data may be shared upon reasonable request to the Ethics Committee at xwkyll@xwh.ccmu.edu.cn.

**Funding:** The author(s) received no specific funding for this work.

**Competing interests:** The authors have declared that no competing interests exist.

significant reduction in risk-adjusted mortality over time compared to younger patients (elderly vs. younger, 12.5% vs. 6.3%, relative reduction per year, $p < 0.001$).

## Conclusion

ICU admission rates for elderly patients are declining, particularly in nonsurgical cases. Less invasive life support modalities have been increasingly utilized in their care. While mortality remains higher among elderly patients, they demonstrate a more significant improvement in survival over time compared to younger cohorts.

## Introduction

The global population is aging, and life expectancy continues to rise. In China, the number of individuals aged 80 and over is projected to be 100 million by 2050, 2.7 times more than in 2021 [1]. The acute and chronic illnesses prevalent in this age group are significant drivers of the escalating demand for critical care services. Although a noticeable increase in the admission of elderly patients to critical care units has been reported in countries such as Australia [2], several European nations [3–5], and parts of the UK (excluding Scotland) [6], the trend is not uniformly observed across all developed countries, particularly in areas with limited resources [7,8]. The recent decline in elderly patients admitted has raised concerns about the potential for age-based rationing of critical care access, leading to inequitable treatment opportunities.

Elderly patients are often provided with less intensive and shorter treatments in ICUs compared to younger patients [5,9–11]. Several factors, including concerns about the harmful effects of aggressive treatments on older individuals, fears of futile care, and economic limitations, may contribute to this disparity. However, ongoing advancements in medicine, surgery, and therapeutics, combined with social progress, may influence how ICU resources are utilized and potentially improve outcomes for elderly patients.

We examined the potential impact of resource limitations on admission trends and care practices at a tertiary referral teaching hospital in Beijing, China, from 2014 to 2021. According to the 2021 China Health Statistics Yearbook, China had approximately 4.75 intensive care unit beds per 100,000 people, a figure significantly lower than that in high-income countries [12]. Furthermore, our study was initiated during the COVID-19 outbreak, this project gains additional relevance due to the heightened stress on healthcare resources during the pandemic [13].

This study seeks to analyze and compare trends in ICU admissions, clinical outcomes, and resource use among critically ill elderly patients (≥80 years) in comparison to older (65–79 yr) and younger (16–64 yr) cohorts over an 8-year period (2014–2021).

## Methods

### Study design and population

The study was carried out at a tertiary referral teaching hospital in Beijing, China, which has a total of 1,643 beds, including 128 ICU beds (S1 Table). Retrospective

data were collected from the electronic medical records (EMR) system of adult patients (age ≥ 16 yr) who had their first ICU admission between January 1, 2014 and December 31, 2021. Patients were categorized into three age groups based on widely reported thresholds in the existing literature [14]: elderly (≥80 yr), older (65–79 yr), and younger (<65 yr) at the time of ICU admission. Trends over time and hospital mortality were further analyzed across three subgroups: elective surgical, emergency surgical, and nonsurgical patients.

## Variables

Data collection included the following items: firstly, demographic and clinical profiles of patients at the time of ICU admission, including age, sex, BMI, source of admission, and details of surgical procedures that occurred either on the day of ICU admission or within a 7-day period preceding their admission [5]. Based on these criteria, patients were categorized into surgical and nonsurgical groups. Surgical patients were further divided into elective or emergency based on urgency codes recorded in the Surgical and Anesthetic Record. Primary diagnosis were grouped into eight categories (cardiovascular, respiratory, gastrointestinal, neurologic, infectious, musculoskeletal/injury-related, hematologic/oncologic and other diseases) [15]. Illness severity was evaluated using both the Acute Physiological and Chronic Health Evaluation II (APACHE II) score and Age-corrected APACHE II, calculated by subtracting age points from the APACHE II. Preexisting morbidity was determined based on prior diagnoses of the 19 conditions in the Charlson Comorbidity Index (CCI), calculated using hospital discharge codes recorded in the Home Page of Medical Records. Morbidity was then categorized into low (CCI = 0), moderate (CCI = 1–2), and high (CCI ≥ 3) groups.

Then data on ICU resource use: use of invasive mechanical ventilation, renal replacement therapy (RRT), Inotropics/vasopressors (epinephrine, norepinephrine, dopamine, metaraminol, and dobutamine), nutrition support, and invasive procedures (tracheostomy, urinary catheterization, central venous catheterization, and arterial catheterization). Additional metrics included the duration of mechanical ventilation, length of stay (LOS) in the ICU and hospital, and Therapeutic Intervention Scoring System-28 scores (TISS-28), which summarize therapeutic, diagnostic, and nursing workloads in the ICU. TISS-28 data were recorded daily until ICU discharge. We have presented the sum of TISS divided by the number of days of ICU stay; thus TISS was adjusted for length of stay [16]. In addition, we also paid special attention to the total blood loss for repeated laboratory testing during ICU stay (For a detailed description of variables, see S1 Text.) and red blood cell (RBC) transfusion. Finally, ICU and hospital mortality. Data were complete for all patients.

To contextualize the trends in elderly ICU admissions, we also extracted data on the total number of hospital admissions and the number of admissions for patients aged ≥80 years at the same institution during the same study period (January 1, 2014 to December 31, 2021) from the hospital administrative database. These data were used to calculate the proportion of elderly patients in the overall hospitalized population for comparison with the proportion of elderly patients among ICU admissions.

## Outcomes

The main outcome was hospital mortality, while secondary outcomes included ICU mortality and resource utilization, which was assessed based on treatment intensity (invasive mechanical ventilation, RRT, inotropics/vasopressors, nutrition support, and duration of organs supported); ICU and hospital LOS and TISS-28.

## Ethics

The study was approved by the Xuanwu Hospital Human Research Ethics Committee in Beijing, China (approval number 2024(047)-001). Given the retrospective and de-identified nature of the data, informed consent was waived. Data were accessed from Xuanwu hospital EMR system between May 1, 2024 and October 8, 2024. Ethical approval was valid from April 28, 2024 to April 28, 2025. Authors couldn't identify individual participants during or after data collection.

## Statistical analysis

Data analysis was conducted using SPSS 25.0, STATA 18.0 (Stata, College Station, TX), and R version 4.4.1 (R Foundation for Statistical Computing, 2024). Non-normally distributed variables were compared with the Kruskal-Wallis test, while categorical and ordinal variables were analyzed using the chi-square test and chi-square test for trend, respectively. A p-value <0.05 was considered statistically significant. All significance tests were two–tailed; 95% CIs were presented where appropriate. Continuous variables were summarized as means and SD or medians and IQR. Categorical variables were expressed as n (%).

Multinomial regression analysis was employed to examine unadjusted trends in baseline characteristics, resource utilization, and hospital mortality from 2014 to 2021. Risk-adjusted trends in hospital mortality were assessed using logistic regression models, which adjusted for sex, admission type, Charlson comorbidity index, Acute Physiology Score (APS), and primary diagnosis. Time was treated as a continuous variable, with an interaction term included to test for differences in mortality trends across age groups. Results were reported as odds ratios with 95% CIs. Model performance was evaluated using the area under the receiver-operating characteristic (ROC) curve and the Brier score. Annualized marginal risk-adjusted probabilities were calculated and graphically displayed to illustrate changes in hospital mortality over time.

## Results

### Demographics and baseline characteristics

We identified 31,535 admissions aged ≥ 16 years who were first admitted to the critical care units at XuanWu Hospital in Beijing from 1 January 2014 to 31 December 2021 (Fig 1). The median (IQR [range]) annual number of admissions was 3,663 (3552–4350 [3529–5240]). Summary of patient demographics and baseline characteristics is presented in Table 1. The median age was 62 (51–72) years. While there was a male predominance across the entire cohort, this was relatively less pronounced in the elderly group.

Elderly patients, who accounted for 11.5% of admissions, exhibited a higher rate of admissions from emergency department, along with a greater burden of comorbid diseases and a more severe illness, both with and without age adjustment (APACHE II and Age-corrected APACHE II). Admission patterns varied across age groups. Specifically, the proportion of patients admitted following elective or emergency surgeries declined as age increased. In contrast, a larger proportion of

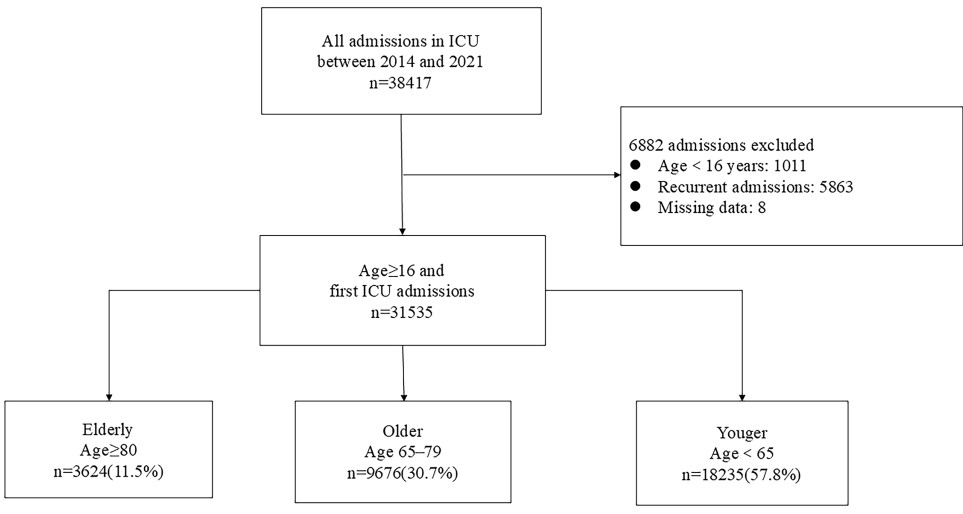

**Fig 1. Study diagram.**

**Table 1. Patient characteristics at intensive care unit admission.**

| | Total | 16–64Yr | 65–79Yr | ≥80Yr | P |
|---|---|---|---|---|---|
| No. of admissions, n(%) | 31535 | 18235(57.8) | 9676(30.7) | 3624(11.5) | |
| Age, median(IQR) | 62(51–72) | 53(43–59) | 71(67–75) | 83(81–86) | <0.001 |
| Male, n(%) | 18573(58.9) | 11074(60.70) | 5578(57.60) | 1921(53.00) | <0.001 |
| BMI, median(IQR) | 24.5(22.1–27.0) | 24.7(22.3–27.2) | 24.5(22.1–26.9) | 23.7(20.8–26.3) | <0.001 |
| Admission from emergency room, n(%) | 18808(59.6) | 10068(55.2) | 5761(59.5) | 2979(82.2) | <0.001 |
| Admission type, n(%) | | | | | <0.001 |
| Elective surgical | 14043(44.5) | 8984(49.3) | 4278(44.2) | 781(21.6) | |
| Emergency surgical | 6701(21.2) | 4290(23.5) | 1868(19.3) | 543(15.0) | |
| Nonsurgical | 10791(34.2) | 4961(27.2) | 3530(36.5) | 2300(63.5) | |
| APACHEII, median(IQR) | 9.0(6.0–14.0) | 7.0(5.0–12.0) | 11.0(9.0–15.0) | 14.0(10.0–18.0) | <0.001 |
| Age-corrected APACHEII, median(IQR) | 6.0(3.0–10.0) | 5.0(3.0–9.0) | 6.0(3.0–10.0) | 8.0(4.0–12.0) | <0.001 |
| Charlson comorbidity index, n(%) | | | | | <0.001 |
| Low(0) | 6665(21.1) | 5168(28.3) | 1148(11.9) | 349(9.6) | |
| Medium(1–2) | 14933(47.4) | 8988(49.3) | 4417(45.6) | 1528(42.2) | |
| High(≥3) | 9937(31.5) | 4079(22.4) | 4111(42.5) | 1747(48.2) | |
| Principal diagnosis, n(%) | | | | | <0.001 |
| Infectious | 1915(6.1) | 889(4.9) | 532(5.5) | 494(13.6) | |
| Cardiovascular | 8932(28.3) | 4232(23.2) | 3234(33.4) | 1466(40.5) | |
| Respiratory | 997(3.2) | 390(2.1) | 324(3.3) | 283(7.8) | |
| Gastrointestinal | 1896(6.0) | 995(5.5) | 516(5.3) | 385(10.6) | |
| Neurologic | 7856(24.9) | 5085(27.9) | 2321(24.0) | 450(12.4) | |
| Hemotologic and oncologic | 8305(26.3) | 5676(31.1) | 2318(24.0) | 311(8.60) | |
| Musculoskeletal and injuries | 914(2.9) | 461(2.5) | 288(3.0) | 165(4.6) | |
| Other diseases | 720(2.3) | 507(2.8) | 143(1.5) | 70(1.9) | |

APACHEII:the Acute Physiological and Chronic Health Evaluation II.

elderly patients were admitted for nonsurgical conditions. Furthermore, they also had higher rates of infectious, respiratory, cardiovascular, gastrointestinal, musculoskeletal and injurie-related diagnoses and lower rates for neurologic, hematologic and oncologic-related admissions compared with younger age patients. During the study period, the proportion of elderly admissions with high comorbidity remained stable. However, APACHE II scores increased significantly across all age groups, with the smallest increase in the elderly (S2 Table).

## Admission trends

Although the total number of patients and ICU beds increased over the study period, the proportion of elderly patients admitted to the ICU showed a significant decline, with a relative decrease of 31.8%, from 13.2% in 2014 to 9.0% in 2021. In contrast, the proportion of younger patients increased from 54.7% in 2014 to 59.7% in 2021, a relative increase of 9.1%, while the proportion of older patients remained stable (Fig 2; S1 Table). Among the elderly, admissions increased in the 85–89 and 90–94 age groups, from 25.6% to 31.6% (p = 0.017) and from 6.0% to 9.5% (p = 0.037), respectively. Conversely, the 80–84 age group saw a decrease from 66.9% to 57.9% (p < 0.001; S1 Fig). Within the elderly group, the decreasing trend was primarily seen in nonsurgical admissions, while both elective and emergency surgical admissions showed an increasing trend (p < 0.001; S2 Fig).

To assess whether the observed decline reflected broader hospitalization patterns or selective triage, we compared the proportion of elderly patients (≥80 years) among all hospital admissions with that among ICU admissions. The proportion

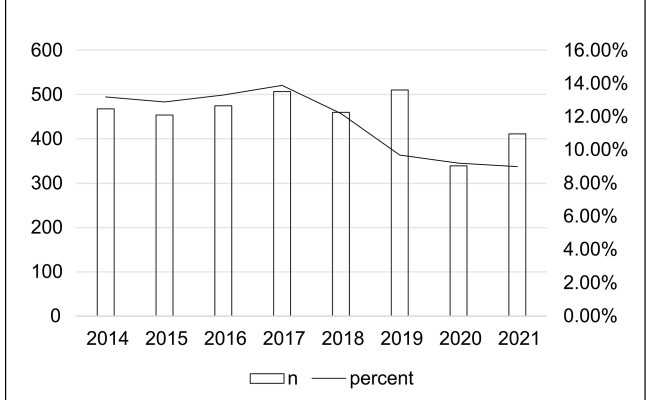

**Fig 2. Number and percentage of elderly ICU patients (≥80 years) over the study period.** The bars represent the annual number of elderly ICU admissions, and the line shows the percentage of elderly patients relative to the overall ICU admissions. Trend in proportion over time, P < 0.001.

in the total hospital population decreased modestly from 6.34% in 2014 to 5.79% in 2021 (relative decrease 8.7%), whereas the decline in elderly ICU admissions was 31.8%—nearly four times greater (S1 Table).

## ICU resource use

The daily average TISS-28 score was negatively correlated with age (P < 0.001), being significantly lower in the elderly group. Compared to younger patients, elderly patients were more likely to receive inotropes/vasopressors and nutritional support, but less likely to undergo invasive ventilation or procedures such as tracheostomy, urinary catheterization, central venous catheterization, and arterial catheterization. They also experienced greater blood loss for laboratory testing and had more frequent red blood cell transfusions and albumin administration. Although they required a longer duration of ventilation and ICU stays, their overall hospital stays were shorter (Table 2).

Over the study period, the proportion of elderly patients requiring invasive ventilation decreased annually, while the proportion receiving enteral nutrition support increased each year. All groups exhibited a significant upward trend in daily average TISS-28 scores. ICU length of stay increased significantly in both the elderly and older groups, while hospital length of stay remained unchanged (S3 Table).

## Mortality

The overall in-hospital mortality rate for all patients was 5.3%, with 14.1% in the elderly group. This rate significantly increased with age, a trend that remained consistent even within further age stratifications among elderly patients (S3 Fig). Additionally, elderly patients with nonsurgical diagnoses had the highest mortality rates (Table 3). In-hospital mortality was associated with male gender, older age, high APS score, high CCI, and nonsurgical conditions (multivariable model in S5 Table).

**Trends in mortality.** Unadjusted hospital mortality decreased significantly in the elderly and older groups, with the elderly mortality rate dropping from 20.7% in 2014 to 10.2% in 2021. A reduction was also seen in elective surgical admissions across all age groups. Among nonsurgical admissions, a decline was noted only in the elderly group (S4 Table). Risk-adjusted hospital mortality decreased across all age groups, with a relative reduction of 12.3% per year for the elderly (95% CI, 8.0%–16.5%), 13.5% for the older group (95% CI, 9.8%–16.9%), and 6.1% for the younger group (P < 0.05) (Fig 3). Summary of multivariable models analyzing adjusted mortality trends is provided in S6 Table.

**Table 2. Resource use in ICU population.**

| | Total | 16–64Yr | 65–79Yr | ≥80Yr | P |
|---|---|---|---|---|---|
| | n = 31535 | n = 18235 | n = 9676 | n = 3624 | |
| Daily average TISS-28, median(IQR) | 22.1(18.0–24.0) | 23.0(18.0–24.3) | 22.0(17.5–24.0) | 20.6(17.0–24.0) | <0.001 |
| Organ support | | | | | |
| Invasive ventilation, n(%) | 8219(26.1) | 5124(28.1) | 2411(24.9) | 684(18.9) | <0.001 |
| Invasive ventilation(>48h), n(%) | 3592(11.3) | 2049(11.2) | 1128(11.7) | 415(11.5) | 0.570 |
| Duration of invasive ventilation(>48h), median(IQR), h | 238.0(123.0–419.0) | 230.0(116.0–403.0) | 247.0(134.0–432.0) | 261.0(141.0–483.0) | 0.001 |
| RRT, n(%) | 535(1.7) | 302(1.7) | 167(1.7) | 66(1.8) | 0.754 |
| Inotropics/vasopressors, n(%) | 2283(18.8) | 1107(16.8) | 795(21.1) | 381(22.6) | <0.001 |
| Enteral nutrition support, n(%) | 6541(20.7) | 3707(20.3) | 2003(20.7) | 831(22.9) | 0.002 |
| Parenteral nutrition, n(%) | 8242(26.1) | 4550(25.0) | 2555(26.4) | 1137(31.4) | <0.001 |
| Invasive Procedures, n(%) | | | | | |
| Tracheostomy | 1462(4.6) | 958(5.3) | 402(4.2) | 102(2.8) | <0.001 |
| Urinary Catheterization | 22637(71.8) | 13561(74.4) | 6797(70.2) | 2279(62.9) | <0.001 |
| Central Venous Catheterization | 8439(26.8) | 5124(28.1) | 2504(25.9) | 811(22.4) | <0.001 |
| Arterial Catheterization | 2640(8.4) | 1533(8.4) | 923(9.5) | 184(5.1) | <0.001 |
| Blood loss for laboratory testing, ml, median(IQR) | 25(16–54) | 22(14–47) | 25(17–53) | 32(20–65) | <0.001 |
| Red cells usage, n(%) | 3883(12.3) | 2061(11.3) | 1194(13.4) | 528(14.6) | <0.001 |
| Albumin Infusion, n(%) | 8223(26.1) | 4385(24.0) | 2717(28.1) | 1121(30.9) | <0.001 |
| ICU LOS, median(IQR), d | 2.8(0.9–6.6) | 2.1(0.8–5.7) | 2.9(0.9–6.8) | 4.7(2.0–9.1) | <0.001 |
| Hospital LOS, median(IQR), d | 12.0(7.9–18.0) | 12.0(7.9–18.0) | 12.5(8.0–18.7) | 11.5(7.5–16.9) | <0.001 |

TISS-28:Therapeutic Intervention Scoring System-28 scores; RRT:renal replacement therapy; LOS:length of stay.

**Table 3. Outcomes in ICU population.**

| | Total | 16–64Yr | 65–79Yr | ≥80Yr | P |
|---|---|---|---|---|---|
| | n = 31535 | n = 18235 | n = 9676 | n = 3624 | |
| **Mortality** | | | | | |
| Hospital mortality, n(%) | 1671(5.3) | 566(3.1) | 594(6.1) | 511(14.1) | <0.001 |
| Elective surgical, n(%) | 191(1.4) | 64(0.7) | 79(1.8) | 48(6.1) | <0.001 |
| Emergency surgical, n(%) | 321(4.8) | 137(3.2) | 116(6.2) | 68(12.5) | <0.001 |
| Nonsurgical, n(%) | 1159(10.7) | 365(7.4) | 399(11.3) | 395(17.2) | <0.001 |
| ICU mortality, n(%) | 1482(4.7) | 527(2.9) | 520(5.4) | 435(12.0) | <0.001 |
| **ADL(survived)** | | | | | |
| ADL at admission, median(IQR) | 70(40–100) | 85(40–100) | 70(40–95) | 45(25–65) | <0.001 |
| ADL at discharge, median(IQR) | 85(55–95) | 90(60–100) | 85(50–95) | 65(35–85) | <0.001 |
| ΔADL(discharge score – admission score), median(IQR) | 0(−10–20) | 0(−10–20) | 0(−10–20) | 10(0–25) | <0.001 |

ADL:Barthel Index for Activities of Daily Living.

## Discussion

Between 2014 and 2021, elderly patients accounted for 11.5% of ICU admissions, but this proportion decreased annually, particularly among nonsurgical patients. Elderly patients exhibited a higher comorbidity burden and a greater severity of illness. Their resource utilization was characterized by a low incidence of invasive mechanical ventilation, which continued to decline yearly. On the other hand, there was a high rate of relatively non-invasive organ support, such as higher use

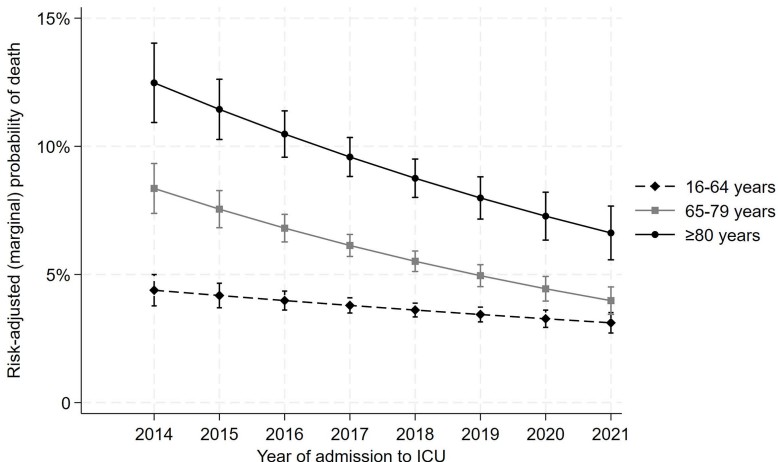

**Fig 3. Mortality risk change over time.** The figure illustrates the predicted (marginal) mortality risk based on a regression model (ICU admission date as a continuous variable, showing the change in the odds of death per year, with an interaction term to compare differences among the three age groups). P value for the difference in slopes between groups was < 0.05.

of notropic/vasopressors and nutritional support. Despite their higher mortality rates, elderly patients showed a greater improvement in survival over time compared to their younger counterparts.

Despite projections indicating a growing elderly population, our study found a significant decline in the proportion of elderly ICU admissions. This contrasts with reports of rising trends in ICU admissions among those aged ≥80 years in other countries, including Australia (5.6% increase per year, 2000–2005) [2], the Republic of Korea (increase from 8.6% to 13.6%, 2002–2010) [17], and Denmark (increase from 11.7% to 13.8%, 2005–2011) [5]. Differences in admission policies, such as a lower admission threshold for the elderly, or expansions in critical care bed capacity may explain these discrepancies [5,18]. Instead, our results align more closely with trends observed in Scotland and Wales [7,14]. In Scotland, Docherty et al. reported a 22% relative decrease in ICU admissions for those aged ≥80 years, and a 16% decrease for those aged 65–79 years between 2005 and 2009, raising concerns about age-based rationing and inequitable access to critical care [7]. In Wales, a similar decline in ICU admissions for patients aged ≥65 years was observed relative to the national population from 2008 to 2017, accompanied by a marked reduction in the number of patients with comorbidities and medical diagnoses [14].

Several factors rooted in China's healthcare context likely contribute to this decline. Critical care resources are limited, with only 4.75 ICU beds per 100,000 population—far below high-income countries [12]. Such scarcity may necessitate selective admission, especially for patients perceived to have poorer prognoses. The COVID-19 pandemic (2020–2021) further strained bed capacity, potentially intensifying this selectivity. To test whether the decline simply mirrors broader hospitalization patterns, we compared the proportion of elderly patients in ICU admissions with that in all hospital admissions. The proportion in the overall hospital population fell only modestly (relative decrease 8.7%), whereas the decline in elderly ICU admissions was 31.8%—nearly four times greater. This disproportionate reduction cannot be explained by changes in the underlying hospitalized elderly population and instead supports the possibility of more selective triage of the oldest old during resource scarcity, consistent with concerns raised by Docherty et al. [7] regarding age-based rationing.

While overall elderly ICU admissions decreased, the share of patients aged ≥85 years increased. This very elderly subgroup warrants particular attention, as age alone is an insufficient criterion for resource allocation [16,19,20], yet both short- and long-term mortality rates remain high [21,22]. Additionally, the decline was driven primarily by nonsurgical

admissions, whereas both elective and emergency surgical admissions showed increasing trends. In line with Docherty et al.'s findings [7], surgical conditions generally yield better outcomes than medical conditions (this was demonstrated by their finding that one-year survival rate among the oldest cohort was 45% for those undergoing emergency abdominal surgery, compared to 25% for those admitted with pneumonia). Similar conclusions have been observed in other studies [23,24]. In our study, elderly patients admitted with nonsurgical diagnoses had the highest mortality rates (17.2% for non-surgical, 12.5% for emergency surgery, and 6.1% for elective surgery). Notably, advances in perioperative management during the study period—including the implementation of Enhanced Recovery After Surgery (ERAS) programs in China since 2016, alongside innovations in multimodal analgesia, goal-directed fluid therapy, and the dissemination of geriatric anesthesia guidelines—have progressively improved surgical safety and feasibility for older patients. Collectively, these developments may have expanded the pool of elderly candidates for elective and even emergency surgery, thereby contributing to the observed increase in surgical ICU admissions.

A closer examination of resource utilization across age groups reveals a seemingly paradoxical pattern: elderly patients, despite higher comorbidity burdens and greater illness severity, received the lowest intensity of care as measured by daily TISS-28 scores. This was primarily driven by a significantly lower use of invasive mechanical ventilation—a trend that continued to decline annually—alongside reduced rates of tracheostomy and other invasive procedures. In contrast, this group exhibited a higher reliance on non-invasive organ support, including more frequent administration of inotropes and vasopressors, enteral and parenteral nutrition, and a greater need for red blood cell transfusions. This pattern may be further contextualized by patient preferences: evidence from multiple settings suggests that a substantial proportion of elderly patients prioritize comfort and quality of life over maximal life extension, and such preferences may influence shared decision-making regarding the intensity of invasive interventions.

This pattern should not be interpreted as therapeutic nihilism or under-treatment. Rather, it reflects a nuanced, physiology-based clinical strategy increasingly adopted in the care of critically ill older adults—one that embodies the principle of "less is more." Invasive interventions such as prolonged mechanical ventilation or tracheostomy carry substantial risks in this population, including ventilator-associated pneumonia, delirium, muscle wasting, and prolonged immobility, all of which may disproportionately affect patients with limited physiological reserve. The deliberate avoidance of such interventions is therefore a proactive strategy aimed at minimizing iatrogenic harm and preserving functional status and dignity, particularly in those with frailty or limited life expectancy. This finding aligns with previous reports showing that organ support, primarily ventilator support and renal replacement therapy, was lower in elderly patients compared to matched younger cohorts [25–27].

Conversely, the increased use of cardiovascular support is consistent with the age-related decline in cardiovascular reserve, which renders elderly patients more vulnerable to hemodynamic instability during acute stress, such as infection or surgery, as observed by Docherty et al. [7]. Similarly, the emphasis on nutritional support addresses the high prevalence of malnutrition and sarcopenia in this cohort, both of which are critical determinants of survival and recovery. The higher rate of red blood cell transfusion observed in elderly patients is likely multifactorial. While laboratory monitoring may contribute to cumulative blood loss, it is important to note that baseline anemia is more common in older adults. Furthermore, elderly patients frequently present with comorbidities such as coronary artery disease or heart failure, conditions that confer lower tolerance to anemia and may necessitate more liberal transfusion strategies to prevent end-organ ischemia or decompensation. Additionally, the heterogeneity in primary diagnoses across groups—such as differences in the proportion of gastrointestinal bleeding or perioperative blood loss—limits our ability to isolate the precise drivers. As such, this finding should be interpreted with caution and highlights the need for more granular data in future studies.

Collectively, these findings suggest a paradigm shift in the delivery of critical care for the elderly: moving away from a one-size-fits-all approach of maximal intervention toward a more individualized strategy that prioritizes the alignment of treatment intensity with physiological reserve, patient values, and anticipated outcomes. This approach, grounded in a deeper understanding of geriatric pathophysiology, represents a more deliberate and patient-centered model of intensive care.

The crude risk of ICU and hospital mortality was higher in elderly patients compared to the two younger age groups, even after adjusting for potential confounders. While these mortality rates are consistent with the existing literature, there is notable variability. Our data show lower hospital mortality rates compared to recent studies of elderly ICU populations [6,28]. A recent systematic review [28] could not account for the variations in mortality among elderly ICU patients across continents (America and Europe). These may be attributed to factors such as differing age thresholds for defining elderly populations [6,21,28,29], variations in ICU admission criteria for elderly patients [29,30] or differences in case mix [20,31,32]. For example, our study had a lower APACHE II score, fewer patients with high comorbidities, and lower rates of mechanical ventilation and renal replacement therapy compared to recent UK registry data [6].

Our study revealed a declining mortality trend across all age groups, particularly among the elderly and older patients. Several studies have also noted improvements in mortality outcomes following critical illness [6,14,33]. Subgroup analyses of hospital mortality trends show that the mortality rate in the elective surgery cohort significantly decreased across all age categories. Notably, in the nonsurgical cohort, a significant reduction in hospital mortality was observed only among elderly patients. Over the eight years of our study, the proportion of elderly patients with severe comorbidities remained stable, while APACHE II scores in this group demonstrated a clear upward trend. Furthermore, elderly patients increasingly utilized less invasive life support modalities. Despite these challenges, significant improvements in mortality rates were still observed in this cohort. We hypothesize that these survival gains are likely attributed to a deeper understanding of underlying disease processes, advancements in healthcare delivery, and the improved organization of critical care services. Specifically, the avoidance of potentially harmful practices may have played a critical role in enhancing patient outcomes.

Our study has several limitations. First, it was conducted at a single center in China, limiting the generalizability of our findings. Therefore, the results may be most representative of large, urban, academic medical centers in China and should be extrapolated to other settings with caution. Nevertheless, the primary focus of this study was on temporal trends in ICU practices rather than the practices themselves, and these observed patterns may offer relevant insights for other ICUs facing similar demographic and resource constraints. Second, long-term outcomes such as long-term mortality, functional and cognitive recovery, and healthcare resource utilization were not systematically recorded in the ICU database, which limits our ability to provide comprehensive long-term analyses. While short-term mortality is an important metric for evaluating the impact of intensive care, the absence of long-term data limits the practical recommendations we can make. Lastly, additional limitations include the observational nature of the data and the absence of discharge destination information. Despite these, the database provides a valuable perspective for observing trends relevant to healthcare resource planning. Furthermore, documenting and understanding pandemic-related trends offers significant value, enabling a deeper understanding of the impact of the COVID-19 pandemic on elderly population.

## Conclusions

The proportion of elderly ICU admissions declined annually, particularly for nonsurgical cases. Despite higher comorbidity burdens and illness severity, these patients received less invasive life support and showed greater improvements in survival over time compared with younger cohorts.

For clinicians and policymakers, these findings offer two key takeaways. First, the declining ICU admission rate for the oldest old—especially during the pandemic—raises important questions about equitable access to critical care during resource scarcity, underscoring the urgent need for evidence-based triage protocols that extend beyond chronological age. Second, the distinct pattern of care received by elderly survivors—less invasive ventilation but more cardiovascular and nutritional support—suggests that a "one-size-fits-all" aggressive approach may be inappropriate. Optimizing outcomes for this growing population may require a paradigm shift toward a more tailored strategy that balances life-sustaining interventions with the avoidance of iatrogenic harm.

## Supporting information

**S1 Table. Number of admissions and ICU beds over the study period.**
(DOCX)

**S2 Table. Trends in ICU Population Characteristics.**
(DOCX)

**S3 Table. Trends in ICU Resource Use.**
(DOCX)

**S4 Table. Trends in Hospital Mortality.**
(DOCX)

**S5 Table. Multivariable logistic regression model for hospital mortality.**
(DOCX)

**S6 Table. Summary of models examining adjusted hospital mortality over time, the interaction between age groups and time, and model performance characteristics.**
(DOCX)

**S1 Fig. Annual admission demographic trends of the elderly age subgroups across the study period.**
(TIF)

**S2 Fig. Rates of admission for nonsurgical, emergency and elective surgical treatment over time in the elderly group.**
(TIF)

**S3 Fig. Hospital mortality rates across the elderly age subgroups.**
(TIF)

**S1 Text. Definition of blood loss from laboratory testing.**
(DOCX)

## Author contributions

**Conceptualization:** Xiaohui Zhu.

**Data curation:** Xiaohui Zhu, Meiping Wang, Yawei Guo, Rong Sun, Li Jiang.

**Methodology:** Xiaohui Zhu, Meiping Wang, Yawei Guo.

**Resources:** Li Jiang.

**Supervision:** Li Jiang.

**Validation:** Meiping Wang, Yawei Guo.

**Writing – original draft:** Xiaohui Zhu.

**Writing – review & editing:** Xiaohui Zhu, Li Jiang.

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
