## [Decision Letter · Decision Letter 0]

17 Mar 2026

PONE-D-25-21611Trends in admission, resource use and outcomes among elderly patients admitted to an intensive care unit in ChinaPLOS One 

Dear Dr. Jiang,

Thank you for submitting your manuscript to PLOS ONE. After careful consideration, we feel that it has merit but does not fully meet PLOS ONE’s publication criteria as it currently stands. Therefore, we invite you to submit a revised version of the manuscript that addresses the points raised during the review process.

the Authors etrospectively analyzed ICU patients aged ≥16 years admitted to a tertiary referral

hospital in China from January 2014 to December 2021. A total of 31,535 patients

were categorized into three age groups: ≥80 years (11.5%), 65–79 years (30.7%), and

<65 years (57.8%). The Authors concluded that ICU admission rates for elderly patients are declining, particularly in nonsurgical cases.

Less invasive life support modalities have been increasingly utilized in their care.

The topic is interesting and the paper is well written.

**However major issues can be raised:**

a. The decline in ICU admission rates should be further discussed

b. Higher comorbidities and less invasive support modalities deserve a more n-depth discussion

c. As the Authors stated, their results cannot be generalized.

d. We suggest to add a paragraph, describing the take-home messages.

We look forward to receiving your revised manuscript.

Kind regards,

Chiara Lazzeri

Academic Editor

PLOS One

**Journal Requirements:**

1. When submitting your revision, we need you to address these additional requirements. Please ensure that your manuscript meets PLOS ONE's style requirements, including those for file naming. The PLOS ONE style templates can be found at https://journals.plos.org/plosone/s/file?id=wjVg/PLOSOne_formatting_sample_main_body.pdf and https://journals.plos.org/plosone/s/file?id=ba62/PLOSOne_formatting_sample_title_authors_affiliations.pdf 2. We note that you have indicated that there are restrictions to data sharing for this study. For studies involving human research participant data or other sensitive data, we encourage authors to share de-identified or anonymized data. However, when data cannot be publicly shared for ethical reasons, we allow authors to make their data sets available upon request. For information on unacceptable data access restrictions, please see http://journals.plos.org/plosone/s/data-availability#loc-unacceptable-data-access-restrictions.  Before we proceed with your manuscript, please address the following prompts: a) If there are ethical or legal restrictions on sharing a de-identified data set, please explain them in detail (e.g., data contain potentially identifying or sensitive patient information, data are owned by a third-party organization, etc.) and who has imposed them (e.g., a Research Ethics Committee or Institutional Review Board, etc.). Please also provide contact information for a data access committee, ethics committee, or other institutional body to which data requests may be sent. b) If there are no restrictions, please upload the minimal anonymized data set necessary to replicate your study findings to a stable, public repository and provide us with the relevant URLs, DOIs, or accession numbers. Please see http://www.bmj.com/content/340/bmj.c181.long for guidelines on how to de-identify and prepare clinical data for publication. For a list of recommended repositories, please see https://journals.plos.org/plosone/s/recommended-repositories. You also have the option of uploading the data as Supporting Information files, but we would recommend depositing data directly to a data repository if possible. Please update your Data Availability statement in the submission form accordingly. 3. If the reviewer comments include a recommendation to cite specific previously published works, please review and evaluate these publications to determine whether they are relevant and should be cited. There is no requirement to cite these works unless the editor has indicated otherwise.

Reviewers' comments:

Reviewer's Responses to Questions

**Comments to the Author**

1. Is the manuscript technically sound, and do the data support the conclusions?

Reviewer #1: Partly

2. Has the statistical analysis been performed appropriately and rigorously?

Reviewer #1: Yes

3. Have the authors made all data underlying the findings in their manuscript fully available?

Reviewer #1: Yes

4. Is the manuscript presented in an intelligible fashion and written in standard English?

Reviewer #1: Yes

5. Review Comments to the Author

**Reviewer #1:** The trends of elderly patients admitted to an ICU during the years of the COVID-19 pandemic should be compared to the trends in hospital admissions during the same period for the same population. During this specific period of ICU beds shortage, the admission to the ICU could have been denied based on probability of survival and not because they didn’t meet an admission criteria.

6. PLOS authors have the option to publish the peer review history of their article (what does this mean?). If published, this will include your full peer review and any attached files.

**Do you want your identity to be public for this peer review?** For information about this choice, including consent withdrawal, please see our Privacy Policy.

Reviewer #1: No

You may also use PLOS’s free figure tool, NAAS, to help you prepare publication quality figures: https://journals.plos.org/plosone/s/figures#loc-tools-for-figure-preparation

---

## [Author Response · Author response to Decision Letter 1]

18 Apr 2026

To: The Academic Editor, PLOS ONE

Manuscript ID: PONE-D-25-21611

Title: Trends in admission, resource use and outcomes among elderly patients admitted to an intensive care unit in China

Authors: Xiaohui Zhu, Meiping Wang, Yawei Guo, Rong Sun, Li Jiang

Dear Dr. Lazzeri and Reviewer,

Thank you very much for your thoughtful and constructive feedback on our manuscript. We greatly appreciate the opportunity to revise our work. We have carefully addressed all the comments raised by the Academic Editor and the Reviewer. Below we provide a point-by-point response detailing the specific revisions made. All changes are highlighted in the marked-up revised manuscript.

Response to Academic Editor’s Comments

General comment: The topic is interesting and the paper is well written. However major issues can be raised: a) The decline in ICU admission rates should be further discussed; b) Higher comorbidities and less invasive support modalities deserve a more in-depth discussion; c) As the authors stated, their results cannot be generalized; d) We suggest to add a paragraph, describing the take-home messages.

Response:

We thank the Editor for the encouraging assessment and for the specific, actionable suggestions. We have substantially revised the manuscript to address each point:

a) Further discussion of the decline in ICU admission rates

We have expanded the Discussion to provide a deeper, multi-faceted analysis of the observed decline in elderly ICU admissions (≥80 years). Specifically:

• We now explicitly compare the decline in elderly ICU admissions (relative decrease 31.8%) with the change in the proportion of elderly patients in the overall hospital population during the same period (relative decrease 8.7%). This comparison, presented in the Results section (page 7, lines197–202) and supported by S1 Table, demonstrates that the decline is nearly four times greater than what would be expected from broader hospitalization trends, supporting the interpretation of selective triage.

• We discuss the role of China’s limited critical care resources (4.75 ICU beds per 100,000 population) and the additional strain imposed by the COVID 19 pandemic (2020–2021) as key contextual factors that may have intensified admission selectivity (Discussion, page 11, lines 275–280).

• We also discuss the increase in the share of patients aged ≥85 years within the elderly ICU cohort, and the divergent trends between nonsurgical (declining) and surgical (increasing) admissions, linking these to advances in perioperative management(Discussion, pages 11).

b) In-depth discussion of higher comorbidities and less invasive support modalities

We have substantially rewritten the Discussion to provide a more nuanced interpretation of the seemingly paradoxical pattern: elderly patients had higher comorbidity burden and illness severity yet received lower-intensity care as measured by daily TISS-28 scores, with significantly lower use of invasive mechanical ventilation and higher reliance on non invasive organ support (inotropes/vasopressors, nutritional support, RBC transfusions).

• We now argue that this pattern reflects a deliberate, physiology based clinical strategy embodying the principle of “less is more,” aimed at minimizing iatrogenic harm (ventilator associated pneumonia, delirium, muscle wasting) in patients with limited physiological reserve (Discussion, pages 12–13).

• This pattern may be further contextualized by patient preferences: evidence from multiple settings suggests that a substantial proportion of elderly patients prioritize comfort and quality of life over maximal life extension, and such preferences may influence shared decision-making regarding the intensity of invasive interventions(page 12).

• We discuss the increased use of cardiovascular support in relation to age related decline in cardiovascular reserve, and the emphasis on nutritional support as a response to high prevalence of malnutrition and sarcopenia (page 12).

• We also acknowledge the multifactorial nature of higher RBC transfusion rates, including baseline anemia, lower anemia tolerance due to comorbidities (e.g., coronary artery disease, heart failure), and potential contribution from laboratory testing (page 12–13).

c) Generalizability

We have revised the Limitations section to more explicitly address generalizability. We now state that the findings are most representative of large, urban, academic medical centers in China and should be extrapolated to other settings with caution. However, we emphasize that the primary focus was on temporal trends in ICU practices, which may offer relevant insights for other ICUs facing similar demographic and resource constraints (Limitations, page 14, lines384–390).

d) Take home messages paragraph

We have added a dedicated paragraph at the end of the Conclusions section (page 14–15, lines407–416) summarizing two key takeaways for clinicians and policymakers:

1. Equitable access during resource scarcity– The disproportionate decline in elderly ICU admissions (especially during the pandemic) underscores the need for evidence based triage protocols that go beyond chronological age.

2. Tailored, less invasive care– The distinct pattern of care received by elderly survivors suggests that a “one size fits all” aggressive approach may be inappropriate; optimizing outcomes requires a paradigm shift toward a strategy that balances life sustaining interventions with avoidance of iatrogenic harm.

These additions directly address the Editor’s request for a paragraph of take home messages.

Response to Reviewer 1

Reviewer’s comment: The trends of elderly patients admitted to an ICU during the years of the COVID-19 pandemic should be compared to the trends in hospital admissions during the same period for the same population. During this specific period of ICU beds shortage, the admission to the ICU could have been denied based on probability of survival and not because they didn't meet an admission criteria.

Response:

We thank the Reviewer for this important and insightful comment. We fully agree that the COVID 19 pandemic period (2020–2021) introduced unprecedented strain on ICU bed capacity, which could have altered admission decisions for elderly patients. To address this, we have taken the following actions:

• New data extraction and analysis(added to page 4, lines 119–125): We extracted data on the total number of hospital admissions and the number of admissions for patients aged ≥80 years at the same institution during the entire study period (2014–2021) from the hospital administrative database. We then calculated the proportion of elderly patients among all hospital admissions and compared it with the proportion among ICU admissions over time.

• Results (added to page 7, lines 197–202, and S1 Table): The proportion of elderly patients in the overall hospital population decreased modestly from 6.34% in 2014 to 5.79% in 2021 (relative decrease 8.7%). In contrast, the proportion of elderly ICU admissions decreased by 31.8% over the same period—nearly four times greater. This disproportionate reduction cannot be explained by changes in the underlying hospitalized elderly population.

• Discussion (added to page 11, lines 280–288): We now explicitly discuss that this disproportionate decline supports the possibility of more selective triage of the oldest old during resource scarcity, consistent with concerns raised by Docherty et al. (2016) regarding age based rationing. We also note that the pandemic years (2020–2021) saw the lowest proportions of elderly ICU admissions (9.8% and 9.0%, respectively), coinciding with peak bed shortages.

These revisions directly address the Reviewer’s concern by providing a quantitative comparison with hospital wide elderly admission trends and offering a more nuanced interpretation of the role of resource constraints, including the pandemic period.

Response to Journal Requirements

Requirement 1 – PLOS ONE style requirements:

We have carefully reviewed the PLOS ONE style templates and reformatted the manuscript accordingly. This includes adjusting the title page, abstract structure, section headings, reference formatting, and file naming. We confirm that the manuscript now meets the journal’s style requirements.

Requirement 2 – Data availability:

We have revised the Data Availability Statement in the submission form and in the manuscript (end of main text). Due to ethical and legal restrictions imposed by the Xuanwu Hospital Human Research Ethics Committee (approval number 2024(047)-001), the identified data cannot be publicly deposited because they contain potentially identifying patient information (e.g., detailed ICU admission dates, age distributions in small subgroups, and specific diagnostic codes) that could compromise patient privacy even after anonymization. The Ethics Committee imposes this restriction to ensure compliance with China’s regulations on health data protection. Researchers who meet the criteria for access to confidential data may submit requests to the corresponding author (Dr. Li Jiang, jiangli@ccmu.edu.cn), and such requests will be reviewed by the Ethics Committee (xwkyll@xwh.ccmu.edu.cn) for approval.

Summary of Major Revisions

• Introduction: No major changes (already met journal style).

• Methods: Added extraction and analysis of hospital wide admission data for comparison with ICU admission trends (Section “Variables”, page 6).

• Results: Added new comparison of elderly proportion in overall hospital admissions vs. ICU admissions (page 10); updated S1 Table accordingly.

• Discussion: Substantially expanded to include: (a) deeper analysis of decline in elderly ICU admissions with comparison to hospital wide trends; (b) in depth interpretation of higher comorbidities and less invasive support (including “less is more” paradigm, cardiovascular reserve, nutritional support, RBC transfusion); (c) explicit discussion of generalizability; (d) new take home messages paragraph in Conclusions.

• Limitations: Revised to more clearly state generalizability constraints.

• Conclusions: Added dedicated take home messages paragraph.

• Supporting Information: Updated S1 Table with hospital wide admission data.

We believe these revisions have thoroughly addressed all comments raised by the Academic Editor and the Reviewer, and have significantly strengthened the clarity, rigor, and practical relevance of the manuscript. We are grateful for the constructive feedback and hope that the revised manuscript is now suitable for publication in PLOS ONE.

Sincerely,

Li Jiang, MD, PhD

Corresponding Author

Xuanwu Hospital, Capital Medical University

Beijing, China

Email: jiangli@ccmu.edu.cn

---

## [Editor Report · Decision Letter 1]

21 Apr 2026

Trends in admission, resource use and outcomes among elderly patients admitted to an intensive care unit in China

PONE-D-25-21611R1

Dear Dr. Jiang,

We’re pleased to inform you that your manuscript has been judged scientifically suitable for publication and will be formally accepted for publication once it meets all outstanding technical requirements.

Kind regards,

Chiara Lazzeri

Academic Editor

PLOS One
---

## [Editor Report · Acceptance letter]

PONE-D-25-21611R1

PLOS One

Dear Dr. Jiang,

I'm pleased to inform you that your manuscript has been deemed suitable for publication in PLOS One. Congratulations! Your manuscript is now being handed over to our production team.

Kind regards,

on behalf of

Dr. Chiara Lazzeri

Academic Editor

PLOS One